# The Primary Role of Noncoding RNA in the Pathogenesis of Cancer

**DOI:** 10.3390/genes16070771

**Published:** 2025-06-30

**Authors:** Amil Shah

**Affiliations:** Department of Medicine, University of British Columbia, Vancouver, BC V5Z1M9, Canada; amil.shah@ubc.ca

**Keywords:** tumorigenesis, noncoding RNA, RNA regulatory network, cancer attractors, cancer genes

## Abstract

The discovery of oncogenes and tumor suppressor genes provided important insights into the molecular pathogenesis of cancer but also revealed some contradictions in the prevailing somatic mutation theory. The evidence that noncoding RNAs (ncRNAs) form an elaborate network that regulates the flow of genetic information in eukaryotic cells offers an explanation for the inconsistencies. ncRNAs comprise a wide variety of molecules that interact with one another as well as with other RNAs, DNA, and proteins, over whose activities they exert a regulatory influence. The outcome of the dynamic interactions of the cell’s biomolecules is the emergence of higher-order states of equilibrium, called attractor states, which correspond to the gene-expression configurations of distinct cell types. Attractor states are relatively stable systems, but they are susceptible to perturbation by a disturbing force, such as mutation. Mutations that disrupt the ncRNA network can enable the cell to undergo a state transition towards a potentially neoplastic one. This is the crux of tumorigenesis. An inquiry into the architecture of the ncRNA network and its role in tumorigenesis is required to complement our knowledge of the well-known cancer genes as well as serve as a guide in the design of new anticancer therapeutics.

## 1. Introduction

The discovery of oncogenes and tumor suppressor genes in the 1970s reinforced the prevailing somatic mutation theory of tumorigenesis by positing that random, sequential mutations of these genes cumulatively cause large phenotypic shifts towards the hallmarks of cancer: uncoordinated cell proliferation, resistance to apoptosis, clonal expansion, and cell migration to other anatomical sites (metastasis) [1,2]. It was anticipated that a common set of driver mutations would be identified for each cancer type. But this did not turn out to be the case. Among similar cancers in different individuals, there is only modest overlap in gene mutations; in fact, they may not share any mutation in the same gene [3,4,5,6,7]. Conversely, there is an enormous diversity of genetic alterations within the same cancer type between patients to the point that almost every gene in the genome has been associated with malignancy [8]. It has even been postulated that a substantial portion of the somatic mutations in human cancers arise before neoplastic development [9]. At present, there is not a clear picture of precisely which mutations drive tumorigenesis, nor is it always possible to identify the specific biological function that is disrupted [10].

Several recent observations have prompted a rethinking of tumorigenesis [11,12]. These include the puzzling decrease in frequency of some putative driver genes as the cancer evolves, and the lack of congruence in the series of genetic alterations during cancer progression [13,14]. An unexpected finding is that benign conditions, such as nevi or rheumatoid arthritis, harbor cancer genes [15,16]. More surprising is that somatic mutations are prevalent in aging, healthy tissues of individuals without a clear diagnosis of malignancy [17,18,19,20,21,22,23,24]. These observations raise important questions about the exact role of cancer genes in tumorigenesis, benign diseases, and, indeed, normal aging. While there is no debate that oncogenes and tumor suppressor genes contribute to cancer formation, it seems that their impact might be dependent on the coexistent cellular status [25].

Complex biological processes are at play during tumorigenesis, and the patterns of mutational signatures within distinct tumor types could well hold the answer to their causation [26]. In this regard, two avenues of inquiry are salient. First, the dysregulation of gene activities might be more significant to cancer development than aberrant signaling pathways. Here, regulatory noncoding RNAs (ncRNAs) have emerged as crucial elements of an elaborate network that coordinates and modulates gene activities. Second, the dynamic interactions of the cell’s biomolecules (DNA, RNAs, and proteins) collectively establish the gene-expression configurations that specify individual cell types. This article examines the implications of these two notions. Specifically, the perspective put forward is that modifications of the regulatory ncRNA network alter cell state dynamics, causing the cell to veer towards a neoplastic phenotype: this is the underpinning of tumorigenesis.

## 2. The Rise in Noncoding RNAs

### 2.1. From Junk DNA to Key Regulators

The seminal discovery of the double helical structure of DNA, in 1953, by Watson and Crick led to the central dogma of molecular biology: genetic information encoded in DNA is transcribed into RNA as messenger RNA (mRNA) and then translated into protein [27]. In the late 1940s, it was recognized that the amount of genomic DNA in members of a species, called the C-value, is almost identical [28]. However, it was realized that there is a marked disparity in the C-values among different species that does not correspond with their perceived complexity; this is referred to as the C-value paradox [29]. The unexpected finding that a significant proportion of the eukaryotic genome contains DNA that does not code for protein—originally regarded as “junk” DNA—at first seemed to explain the C-value paradox. It was assumed that when the junk DNA was removed, the remaining protein-coding DNA, reflecting the number of genes, would have a relationship with the complexity of organisms. However, when a count of the genes in different organisms was performed, it became evident that no such correlation exists [30]. For example, the number of genes in the human cell is estimated to be about 20,000 [31], not dissimilar to that of the simple worm, *Caenorhabditis elegans*, with 19,700 [32]. This lack of correspondence between the number of genes and the complexity of organisms is termed the G-value paradox [33].

The increasing complexity of eukaryotes over time can be attributed to genomic redesign rather than addition of new genes [34]. The implication is that higher eukaryotes are more sophisticated in how their genes are regulated and expressed; there is, for instance, greater transcriptional control, alternative splicing, and post-translational modifications of peptides to expand their diversity. This concept is captured by the I-value, which asserts that the information contained in the genome is a more reliable benchmark of evolutionary progress [33].

A pertinent discovery is the pervasive transcription of the majority of the genome outside the domain of protein-coding genes [35,36]. Among the transcripts are thousands of heterogenous ncRNAs, which originate in introns and intergenic regions, and other transcripts that do not encode proteins [37]. The ncRNAs interact with one another and with other RNAs, DNA, and proteins to control gene expression in metazoans [38,39]. Indeed, the proportion of the genome transcribed into ncRNAs matches the complexity of an organism more than the number of its protein-coding genes [40]. In this regard, it is noteworthy that noncoding DNA accounts for <25% of the genome in prokaryotes, 25–50% in simple eukaryotes, and >50% in more complex fungi, plants, and animals; this increases to an astonishing 98.5% in humans [37].

### 2.2. Types of Noncoding RNAs

ncRNAs are divided into two classes: housekeeping and regulatory ncRNAs. Housekeeping ncRNAs subserve basic cell functions and are constitutively expressed in relatively stable concentrations across different cell types. First, ribosomal RNA (rRNA), the primary component of ribosomes, makes up the bulk of housekeeping ncRNAs. Second, transfer RNAs (tRNAs) act as intermediaries to carry the appropriate amino acids to the ribosome, based on the mRNA nucleotide sequence. Third, small nuclear RNAs (snRNAs) process pre-messenger RNAs in the nucleus. Finally, small nucleolar RNAs (snoRNAs) chemically modify other RNAs (rRNA, tRNA, and mRNA) through methylation or pseudouridylation. It is worth noting that snoRNA levels have recently been demonstrated to be perturbed in malignant tissues, although their specific role in tumorigenesis is not fully elucidated [41]. The regulatory ncRNAs comprise a diverse group with important roles in coordinating gene expression during development and in maintaining cellular homeostasis [39,42]. Unlike housekeeping ncRNAs, their expression is dynamic in response to specific cellular conditions. Regulatory ncRNAs are further subdivided into three broad categories by genomic size and structure: small ncRNAs (<200 nucleotides (nt) long), long ncRNAs (lncRNAs) (>200 nt), and circular RNAs (circRNAs) (<100 nt to 4000 nt). The different ncRNAs and their main functions are summarized in Table 1.

### 2.3. Functions of Regulatory Noncoding RNAs

Regulatory ncRNAs participate in a wide range of cellular functions, chief among which are transcriptional regulation, RNA processing and modification, and mRNA stability and translational modulation. The most common small ncRNAs are microRNAs (miRNAs), single-strand RNA molecules folded into a stem-loop structure [42,43]. They play several roles, depending on their subcellular locations. In the nucleus, miRNAs promote or repress gene transcription by binding to complementary sequences located on promoters or enhancers and recruiting different protein complexes at these sites [44]. Nuclear miRNAs also regulate pri-miRNA maturation as well as influence mRNA processing. In the cytoplasm, they modulate translation of mRNA [45].

lncRNAs participate in diverse cellular processes through their RNA-protein, RNA-DNA and RNA-RNA interactions. Their different modes of action reflect their large size, flexible conformation and different subcellular localizations [42,46]. Those in the nucleus modify chromatin architecture and regulate transcription by recruiting regulatory factors to specific genomic sites. In the cytoplasm, they serve as scaffolds for protein–protein interactions, and act as decoys of proteins and other RNAs. lncRNAs can also base pair with miRNAs to trigger their degradation; this affects the levels of their down-stream target mRNAs [42].

circRNAs are single-strand RNAs generated by the back-splicing of mRNAs to form covalently closed continuous loops [47]. They influence post-transcriptional gene regulation through the sponging of miRNAs. Some circRNAs function as protein sponges or inhibitors, act as scaffolds to bring different proteins into proximity, or recruit proteins to specific subcellular compartments. Interestingly, specific circRNAs may encode proteins, which participate in gene regulation by forming regulatory loops in various signaling pathways [48].

## 3. Noncoding RNAs in Metazoan Development

### 3.1. Contribution of Noncoding RNAs to Multicellular Development

Prokaryotes contain a predominantly protein-based gene regulatory apparatus, whose components display a quadratic increase in number with genome size [49,50]. Such a system cannot be efficiently scaled and its capacity is quickly exceeded. Therefore, it is not adaptable to multicellular organisms, in which an enormous amount of information is required for the coordination of gene expression and developmental pathways. On the other hand, the ncRNA system appears well suited to circumvent this limitation because of the structural, biosynthetic, and functional diversity of its elements [51].

The enabling features of the ncRNA regulatory system include the following: (a) many ncRNAs are processed after transcription into numerous smaller molecules with different targets, greatly expanding their scope [52]; (b) ncRNA biogenesis can be versatile as illustrated by the processing of lncRNA transcripts, which contain local hairpin structures, by Drosha and Dicer to yield mature miRNAs [42]; (c) the relatively small size of ncRNAs in the range of tens to hundreds of nucleotides, which is about two orders of magnitude smaller than the typical protein molecule, allows them to readily diffuse throughout the cell; (d) ncRNAs operate in different subcellular compartments, extending their sphere of influence on cellular processes; (e) the complementarity of sequence-specific binding of regulatory ncRNAs to their targets sharpens their mode of action; (f) ncRNAs interact with each other as well as with mRNAs to expand their regulatory capacity or modulate post-transcriptional activity by inducing conformational changes [53]; (g) as components of miRNA-induced silencing complexes (miRISCs), miRNAs can respond quickly to changes in subcellular environments and dynamically regulate many target mRNAs through interactions with their complementary sequences, called miRNA response elements or MREs [45]; and (h) ncRNAs typically regulate a few targets, unlike proteins which can regulate hundreds—thus, their effects tend to be more focused and precise [54].

### 3.2. Role of Noncoding RNAs in Cell Development

The essence of metazoan development is the creation of different cell types, each of which acquires specialized functions despite sharing the same genome. Following fertilization, the zygotic genome undergoes reprogramming into a transient totipotent state, from which three germ layers (ectoderm, mesoderm, and endoderm) are laid down in bilaterians. In humans, about 250 different cell types, totaling an estimated 4 × 10^13^ cells in the adult, arise through the process of cell differentiation. The fundamental principle behind cell differentiation is the coordination and modulation of all gene loci to generate distinct gene-expression patterns, each of which corresponds to a specific cell type.

The events that unfold during gene transcription underpin the cell differentiation process. Transcription begins with the transfer of information from specified segments of DNA to RNA, and is controlled by lncRNAs, which serve as scaffolds to tether enzyme complexes on the gene promoters. This facilitates demethylation, chromatin decondensation, and the recruitment of transcription factors and transcription factor-DNA binding motif sequences within enhancers and promoters. In the conventional mRNA-coding protein schema of transcription regulation, the binding of a pioneer transcription factor to chromatin in the gene promoter region causes DNA to become exposed [55]. This allows the binding of transcription factors to the exposed DNA, followed by the recruitment of Pol II to initiate transcription. Transcription then occurs bidirectionally, with sense mRNAs and antisense lncRNAs [56]. Recent evidence, however, suggests that the mechanisms for lncRNA production are different from those for mRNA. The transcription of lncRNAs is initiated by unidentified initiation factors and is mainly *trans*-regulated by lncRNAs from other chromosomes [57]. This model also unveils another aspect of lncRNA regulatory action; while it is widely assumed that lncRNAs bind to the adjacent protein-coding genes by complementary sequences, it seems that most lncRNAs *trans*-regulate their targets across chromosomes. It is also worth noting that in addition to lncRNAs, transcription initiation and regulation are influenced by nuclear miRNAs [44] as well as by epigenetic processes, such as DNA methylation and histone modification [58,59].

Following transcription, further layers of control tune the process to ensure the correct production of proteins in each cell type (post-transcriptional regulation). In the cytoplasm, species of miRNAs bind to mRNAs primarily through base-pairing between the miRNA’s “seed” (2nd–7th nucleotides of miRNA molecule) and the miRNA response elements or MREs, which are predominantly located within the 3′ UTRs of the target mRNAs. After binding to MREs, miRNAs induce translation repression and mRNA deadenylation and decapping [42]. miRNA binding sites are also present in other mRNA regions, including 5′ UTR and coding sequence, where they can exert silencing effects on gene expression [45].

The picture emerges of different ncRNA species operating at critical times through embryonic development and in adult cell types as master regulators to fine-tune gene expression at multiple levels. In the nucleus, they promote the activation or inhibition of gene transcription, whereas in the cytoplasm they affect the stability and translation of mRNAs. Mattick, in 2001, put forward the hypothesis that ncRNAs form a parallel regulatory network that processes and relays information for the genome-wide configuration of gene activities necessary for phenotypic variability [60]. This is in contrast to the usual view of the genetic regulatory network as a hierarchical cascade [61]; instead, the cell’s biomolecules interact to produce a rich network of RNA–RNA, RNA–DNA, and RNA–protein connections with feedback loops [42,62,63].

## 4. Role of Noncoding RNAs in Tumorigenesis

Most studies of the cancer genome to date have focused on mutated oncogenes and tumor suppressor genes, and their associated signal transduction pathways. Mutations can also affect the DNA sequences that specify ncRNAs. Because of the immense diversity of their functions, it is not surprising that these mutations are associated with various diseases, including cancer [64,65]. An analysis of RNAseq data of cancers in The Cancer Genome Atlas (TCGA) database indicates that ncRNAs are critical cancer regulators [66]. Moreover, lncRNAs serve as crucial network hubs, adding further evidence of their role in tumorigenesis.

Within the human genome there are many different forms of genetic variants that are only marginally functional or positively selected. Single-nucleotide variants (SNVs) account for more than 90% of these and they represent the major form of genetic polymorphisms, while short insertions/deletions (INDELS) make up a small minority [67]. Most of the variants (62%) are in the intergenic regions, with the remainder (38%) residing in the protein-coding genes. Further, within the genes, over 95% of the variants are in introns with tiny fractions scattered among 3′ UTRs, 5′ UTRs, and the coding regions. Therefore, the majority of SNVs occurs within the noncoding regions of the genome, where ncRNAs are synthesized. The alterations in ncRNAs due to SNVs disrupt their secondary structure and function, and they are implicated in tumorigenesis [68]. While all the details of the effects of SNVs remain to be fully elucidated, it is noteworthy that a recent examination of the Clinical Proteomic Tumor Analysis Consortium (CPTAC) dataset highlights the biological consequences of uncommon pathogenic and common germline variants in cancer genes. Even though the focus is on coding variants and their association with cognate proteins, it underscores the cumulative impact of these variations on various cancer-associated pathways [69]. 

The functional effects of ncRNA mutations are evident at several cellular levels (Table 2). Chromosomal translocations, which are common events in cancers, lead to the formation of aberrant circRNAs, called fusion circRNAs (f-circRNAs), due to the back-splicing of complementary repetitive intronic sequences (Alu elements) [70]. Normally, circRNAs regulate gene expression by sponging miRNAs. Also, specific circRNAs maintain stem cell pluripotency and control stem cell differentiation. Consequently, abnormal circRNA expression causes an imbalance between self-renewal (proliferation) and differentiation. Another common finding in cancer is chromosomal duplication or deletion, which affects the number of copies of the corresponding ncRNAs.

As noted in the previous section, gene transcription is regulated by lncRNAs, and mutations can disrupt this process. SNVs in the *cis*-regulatory sequences in 3′ UTRs disrupt RNA–RNA or RNA–protein interactions, and, thus, affect post-transcription gene expression, whereas those in the 5′ UTRs influence translation [71,72]. Also, the regulation of some cancer-associated genes by miRNAs can be altered by adenosine-to-inosine (A-to-I) RNA editing in 3′ UTRs [73]; these regions often contain alternative polyadenylation signals that allow the generation of multiple isoforms from a single transcript, thereby facilitating the fine-tuning of expression of certain genes in specific tissues. However, 3′ UTR shortening in cancer cells deletes any regulatory components, such as miRNAs, that it may contain [74]. This loss of miRNAs can indirectly lead to oncogene activation.

Several ncRNAs interact with proteins to promote tumorigenesis. For example, lncRNAs and miRNAs regulate cyclin-dependent kinases (CDKs) and their cyclins at different cell-cycle phases, and alterations due to SNVs are linked to tumorigenesis [75]. Further, in most major cancers, ncRNAs serve as cancer drivers by directly or indirectly interacting with oncogenic proteins. As an example, in prostate cancer, cytoplasmic lncRNAs, like CTB-89H12.4 and Taurine Upregulated Gene 1 (TUG1), act as miRNA sponges to down-regulate *PTEN*, which affects the *PI3K/Akt* pathway [76]. Another example is the catalytic role of lncRNAs in maintaining telomere length; lncRNA TERC (telomerase RNA component) is an essential component of telomerase which, along with TERT (telomerase reverse transcriptase), serves as a template and scaffold for telomerase RNP (ribonucleoprotein) [77]. TERC is overexpressed in various cancers, thus enhancing their replicative potential.

The interplay among the ncRNA species and their interactions with chromatin and proteins point to a different mechanism of tumorigenesis. Rather than an accumulation of discrete mutations in oncogenes and tumor suppressor genes, the dysregulation of the complex ncRNA regulatory network due to mutations affecting ncRNAs is the major driver of cancer development (Figure 1). This concept is discussed in the next section.

## 5. Cells as Dynamic Attractors: Implications for Tumorigenesis

The remarkable success of molecular biology over the past several decades has allowed biologists to dissect DNA to its smallest detail, and its encoded messages have, for the most part, been deciphered. The linear sequence of nucleotides, which specifies the order of amino acids in proteins, provided the map to sort out the complex cellular metabolic pathways. But no clear program that explains the morphogenetic transitions occurring during the development of multicellular organisms is evident. It now appears that this vital aspect of metazoan life is carried out by various factors, which are hard-wired in the genome. Together, they form a highly dynamic, parallel-processing system, whose signals are processed in many different pathways simultaneously, and they are nonlinear and modifiable. 

A special feature of a dynamic collection of variables, which interact with and are influenced by one another, is the spontaneous emergence of novel patterns [78]. This property of self-organization allows the system to eventually settle down into higher-order patterns of equilibrium, called attractor states [79]. Although an attractor state is relatively stable, the system is open and susceptible to perturbation, a disturbing force that can potentially push it in a different direction. Because attractor states tend to be robust, they re-establish spontaneously after minor disturbances. But if strong enough, a perturbation can jolt the system out of one attractor state and into another [80].

Kauffman, in 1969, postulated that the gene-expression profile of each cell type corresponds to an attractor state [81]. It is important to note that the number of theoretically possible attractor states is greater than that of cells in physiological states at any given time, and cells with different degrees of biological fitness emerge [79]. Over time, evolutionary processes selected the genetic programs compatible with orderly cell development from the multitude of possibilities. The resulting genomic landscape becomes “canalized” and streamlined. At the same time, pathways to less fit attractors are not easily accessible and are bypassed [82]. An extension of the concept of a cell as an attractor state is that a cancer cell also represents an attractor state; it exists in the genomic space of all possible gene-expression configurations but is not usually occupied [83].

The ncRNAs collectively form a higher-order regulatory network that determines which genes are active at any moment, and under what conditions. By orchestrating the activities of genes, it defines the trajectories of cell development from embryogenesis through adulthood [84,85]. Mutations can rewire the ncRNA network connections and alter the contours of the genomic landscape. This enables state transitions by steering the cell into a different attractor state. Among the alternative states are gene-expression programs that encode a neoplastic phenotype. This is the crux of tumorigenesis.

## 6. Discussion

As cells become more complex over evolutionary time, the amount of genetic information, particularly regulatory information, increases exponentially [50,51]. New regulators are needed for new genes, and some of these new regulators also require regulation to ensure that they are harmoniously integrated into the existing circuitries in the cell. In metazoans, large DNA segments of the genome code for ncRNAs with diverse cellular functions, chief among which are regulatory signals for cell development [84,86]. It is germane to note that the most evolutionarily conserved miRNAs in bilaterian organisms play a role early in embryonic development, while those that evolved specifically in mammals are active at later stages of embryonic development. Moreover, species-specific miRNAs generally function in adult cell types and not during embryonic development [87]. These observations are in consonance with the view that ncRNAs are the architects of multicellular complexity; they coordinate the activities of genes and mold them into distinct gene-expression configurations.

A cancer arises when the normal gene-expression configuration of a cell is perturbed by mutations, pushing it into an alternative cell state. Because attractor states are resistant to minor perturbations, resetting the cell’s genetic program requires multiple alterations, since individual components of the ncRNA network typically exert a small effect on their own. However, with the accumulation of new genetic variants, the threshold of equilibrium is eventually crossed and the cell undergoes a state transition towards a neoplastic phenotype. This heralds the initiation of tumorigenesis. The transformation to a full-blown cancer is driven by alterations in oncogenes and tumor suppressor genes, which subserve major physiological processes, such as the mitotic cell cycle and apoptotic pathways. They provide the initiated cell with a selective growth advantage that leads to clonal expansion.

There is considerable crosstalk between the ncRNA network and protein-based regulatory circuitries, but they remain distinct entities [88]. This gives rise to dual regulatory mechanisms in the cell: an ncRNA network that presides over essential cell activities, including developmental timing, cell fate decision, and homeostasis, and protein-based circuitries that orchestrate core physiological processes, like cell division and genomic maintenance. In summary, in the context of cancer pathogenesis, the conventional thinking that cancer arises from mutations of oncogenes and tumor suppressor genes may be incomplete. Rather, it appears that they are moderated and controlled by ncRNAs, which serve as the primary drivers for cancer.

Finally, it is important to note that the disruption of ncRNA function can cause various other diseases apart from cancer, including metabolic disorders, cardiovascular disease, neurodegenerative conditions, hematopoietic abnormalities, and immune^.^ disorders [64]. Unlike cancer, where ncRNA alterations bring about a fundamental change in phenotype by rewiring the cell’s ncRNA regulatory network, the ncRNAs involved in benign conditions are linked to specific biochemical pathways or circuits.

### The Next Step: Implications for Cancer Therapy

The perspective that cancer arises from a functional error in the cell development process has important ramifications for new drug discovery. Conventional chemotherapy is directed against replicating DNA and the newer molecularly targeted therapy is designed to reverse the effects of the cancer-driving mutations in oncogenes and tumor suppressor genes. Targeted therapy was introduced with high expectations, but even though more than 30 years have passed and more than 500 targeted anticancer drugs have been approved for clinical use, we continue to struggle to effectively control cancer, especially when they are at an advanced stage or are surgically unresectable. The efficacy of targeted anticancer therapy is stymied by the often-rapid emergence of drug resistance with subsequent cancer relapse. Further, contrary to expectations, targeted therapy is not without toxicity. Common side effects include dermatitis, arthralgia, blurring of vision, gastrointestinal upset, bleeding, and hypertension. Less common are heart failure, hypothyroidism, elevated liver enzymes, leucopenia, and pneumonitis [89].

The entry of ncRNAs into the clinic is presently focused on their use as diagnostic biomarkers [90]. However, in response to the need for more effective therapeutic options, it seems an attractive proposition to target ncRNAs, since they are cell- and tissue-specific [91]. Nucleic acid-based therapeutics represent a new class of drugs for the treatment of cancer at the genetic level through complementary base-pairing [92]. Included among the various therapeutic RNA molecules are antisense oligonucleotides, anti-miRNAs, miRNA mimics, miRNA sponges, circRNAs, small interfering RNAs (siRNAs), and short hairpin RNAs (shRNAs).

Nucleic acid-based therapy is not without its challenges. Ensuring efficient delivery of the drugs to solid tumors as well as to sanctuary sites, like the brain, is a priority. Along the same lines, achieving adequate drug levels in specific cells (cancer cells) is critical. Rapid advances are being made in delivery platforms and several nucleic acid-based therapeutics are now in clinical trials for cancer [93]. It is also important to remain cognizant of potential adverse effects, such as cellular off-target effects, systemic mistargeting, or immune activation. Additionally, because multiple nodes in the ncRNA network are likely perturbed in cancer, the containment of aberrant signals might be a demanding prospect. Another concern comes from the far reach of the ncRNA regulatory network within the cell. The dynamic interactions of its elements determine the network’s output and its vulnerability to perturbation. Therefore, interference by a therapeutic molecule could result in unintended reverberations within the network. An in-depth understanding of the complex regulatory roles of the various ncRNAs and an elucidation of the dynamic links and balances among the various components of the ncRNA network are required in order to mitigate against the potential risks.

## Figures and Tables

**Figure 1 genes-16-00771-f001:**
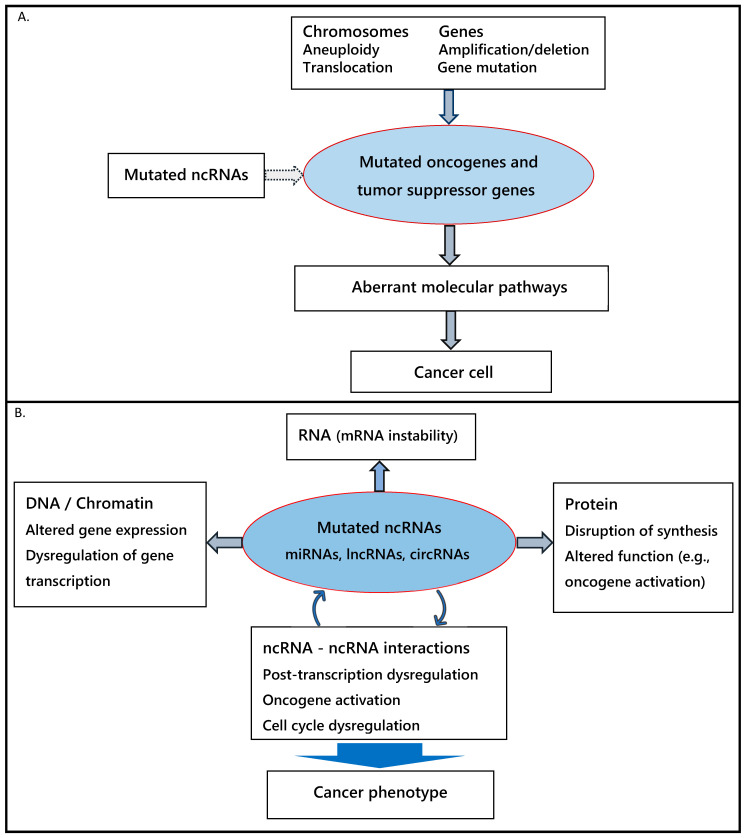
Schematic representation of tumorigenesis models. (**A**) Protein-centric schema, depicting the role of mutated cancer genes, whose abnormal protein products lead to cancer formation through aberrant molecular pathways; ncRNAs interact with these proteins to activate or repress them. (**B**) ncRNA-mediated schema, in which the ncRNA regulatory network is disrupted by mutated ncRNAs; ncRNAs interact with one another as well as with other RNAs, DNA/chromatin, and proteins, collectively leading to a cancer phenotype.

**Table 1 genes-16-00771-t001:** Noncoding RNAs: types and functions.

Noncoding RNAs		
Housekeeping noncoding RNAs		
Type		Main function
Ribosomal RNA (rRNA)		Protein synthesis
Transfer RNA (tRNA)		Protein synthesis
Small nuclear RNA (snRNA)		Pre-mRNA processing
Small nucleolar RNA (snoRNA)		RNA modification
Regulatory noncoding RNAs		
Type		Main function(s)
Small ncRNA (<200 nt)	MicroRNA (miRNA)	Gene transcription (activation/repression) Regulation of pri-miRNA maturation Translational modulation
	Small interfering RNA (siRNA)	Post-transcription regulation
	Piwi-interacting RNA (piRNA)	Silencing of transposons in germ cells
Long ncRNA (lncRNA) (>200 nt)		Spatiotemporal gene expression (cell differentiation) Modification of 3D chromatin architecture Protein scaffolding Decoys of proteins and RNAs miRNA sponging
Circular RNA (circRNA) (<100 nt–4000 nt)		Gene expression (cell/tissue development) miRNA sponging Protein scaffolding

3D: three-dimensional; mRNA: messenger RNA; ncRNA: noncoding RNA; nt: nucleotide.

**Table 2 genes-16-00771-t002:** Examples of the effects of noncoding RNA actions in tumorigenesis.

Target Site/Action	Function/Process Affected
Chromosome	
Translocation (circRNAs)	Gene regulation Cell proliferation–differentiation balance
Chromatin/Gene	
lncRNA-promoter binding	Transcription initiation/regulation
3′ UTRs (SNVs)	Post-transcription gene expression
5′ UTRs (SNVs)	Protein translation regulation
RNA	
mRNA sponging	mRNA degradation
A-to-I editing	Oncogene protein activation
Protein	
ncRNA-protein interaction	Oncogene protein activity
ncRNA-CDK/cyclin interaction	Mitosis
lncRNA TERC- telomerase ribonucleoprotein	Telomere maintenance

3ʹ UTR: 3’ untranslated region; 5’ UTR: 5’ untranslated region; CDK: cyclin-dependent kinase; lncRNA: long noncoding RNA; SNV: single nucleotide variant; TERC: telomerase RNA component.

## Data Availability

No new data were created in this review.

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
