# Peer review of "The Primary Role of Noncoding RNA in the Pathogenesis of Cancer"

_genes, 2025, doi:10.3390/genes16070771_

Round 1
Reviewer 1 Report
Comments and Suggestions for Authors
The manuscript presents a well-organized and insightful exploration of the role of ncRNAs in tumorigenesis, connecting molecular events to systems biology concepts (e.g., attractor states). It effectively challenges the traditional protein-centric model of cancer and provides a strong rationale for considering ncRNAs as both drivers of cancer and targets for therapy. The discussion is conceptually rich and timely, especially with the growing interest in RNA therapeutics. However, there are several areas that would benefit from clarification, additional structure, and visual support.
- The manuscript currently lacks any figures, which significantly reduces its impact and accessibility for readers. The authors could add figures showing the mechanisms by which ncRNA alterations contribute to tumorigenesis. That could be a flowchart or layered diagram showing: SNVs, INDELs and Structural changes in lncRNAs/circRNAs, Effects on gene expression, cell cycle, RNA editing, f-circRNA formation etc. an additional figure could illustrate the attractor model of gene expression and cancer (Normal vs. cancerous attractor states).
- Table 2 is informative but currently dense and hard to read. Ensure consistency in terminology (e.g., SNV vs. SNP), and provide specific examples of each alteration (e.g., name a specific lncRNA affected by SNVs).
- Emphasize the need for system-level integration of RNA data in clinical cancer diagnostics. Point to promising technologies (e.g., single-cell RNA-seq, spatial transcriptomics)
Author Response
Review 1
The manuscript presents a well-organized and insightful exploration of the role of ncRNAs in tumorigenesis, connecting molecular events to systems biology concepts (e.g., attractor states). It effectively challenges the traditional protein-centric model of cancer and provides a strong rationale for considering ncRNAs as both drivers of cancer and targets for therapy. The discussion is conceptually rich and timely, especially with the growing interest in RNA therapeutics. However, there are several areas that would benefit from clarification, additional structure, and visual support.
The manuscript currently lacks any figures, which significantly reduces its impact and accessibility for readers. The authors could add figures showing the mechanisms by which ncRNA alterations contribute to tumorigenesis. That could be a flowchart or layered diagram showing: SNVs, INDELs and Structural changes in lncRNAs/circRNAs, Effects on gene expression, cell cycle, RNA editing, f-circRNA formation etc. an additional figure could illustrate the attractor model of gene expression and cancer (Normal vs. cancerous attractor states).
A new figure (Figure 1, page 7) is added to illustrate the central role of ncRNAs compared with the protein-centric model of tumorigenesis. (I thank the reviewer for this recommendation as it adds to the readability of the paper.)
- Table 2 is informative but currently dense and hard to read. Ensure consistency in terminology (e.g., SNV vs. SNP), and provide specific examples of each alteration (e.g., name a specific lncRNA affected by SNVs).
Table 2 is reformatted for clarity. SNV is used consistently to avoid ambiguity. Examples are added in the manuscript text [Page 9].
- Emphasize the need for system-level integration of RNA data in clinical cancer diagnostics. Point to promising technologies (e.g., single-cell RNA-seq, spatial transcriptomics)
This paper is intended to be an overview of the role of ncRNAs in cancer pathogenesis with a special section [6.1, Pages 9-10] on the importance of this perspective for cancer therapy. A statement is added to highlight the present use of ncRNAs as diagnostic biomarkers [Page 9, lines 447-448]. A follow-up paper that addresses the clinical aspects more fully is now in preparation; this will discuss specific cancers and include the role of ncRNAs in cancer diagnosis and follow-up monitoring for disease relapse.
Reviewer 2 Report
Comments and Suggestions for Authors
The manuscript by Amil Shah provides a comprehensive review discussing the role of ncRNAs in cancer pathogenesis. The author challenges the traditional somatic mutation theory, proposing that dysregulation in the complex ncRNA regulatory network may be a primary driver of tumorigenesis. The article thoroughly discusses the evolution and classification of ncRNAs, the molecular mechanisms by which ncRNAs regulate gene expression, how mutations affecting ncRNAs contribute to cancer development, the concept of cells as "attractor states" and how state transitions driven by ncRNA dysregulation may lead to cancer, as well as potential therapeutic avenues targeting ncRNAs, along with associated challenges. These points are supported with extensive references to recent and relevant literature. The idea that the cumulative rewiring of ncRNA networks leads to neoplastic attractor states represents a conceptual shift, extending beyond gene-centric models toward higher-order regulatory architecture.
While reading the well-written and insightful review, one question came to mind that I would like the author to address: What are the contributions of lncRNA promoter mutation to the described attractor-state model? Transcriptional regulation of lncRNAs themselves is a crucial upstream layer, disruption of which may, e.g., silence tumor-suppressive lncRNAs, activate oncogenic lncRNAs, or create aberrant expression patterns that alter network dynamics. A concise section on lncRNA promoter mutations would strengthen the manuscript by filling an important mechanistic gap and further reinforcing the systems biology argument.
Author Response
Review 2
The manuscript by Amil Shah provides a comprehensive review discussing the role of ncRNAs in cancer pathogenesis. The author challenges the traditional somatic mutation theory, proposing that dysregulation in the complex ncRNA regulatory network may be a primary driver of tumorigenesis. The article thoroughly discusses the evolution and classification of ncRNAs, the molecular mechanisms by which ncRNAs regulate gene expression, how mutations affecting ncRNAs contribute to cancer development, the concept of cells as "attractor states" and how state transitions driven by ncRNA dysregulation may lead to cancer, as well as potential therapeutic avenues targeting ncRNAs, along with associated challenges. These points are supported with extensive references to recent and relevant literature. The idea that the cumulative rewiring of ncRNA networks leads to neoplastic attractor states represents a conceptual shift, extending beyond gene-centric models toward higher-order regulatory architecture.
While reading the well-written and insightful review, one question came to mind that I would like the author to address: What are the contributions of lncRNA promoter mutation to the described attractor-state model? Transcriptional regulation of lncRNAs themselves is a crucial upstream layer, disruption of which may, e.g., silence tumor-suppressive lncRNAs, activate oncogenic lncRNAs, or create aberrant expression patterns that alter network dynamics. A concise section on lncRNA promoter mutations would strengthen the manuscript by filling an important mechanistic gap and further reinforcing the systems biology argument.
This is an insightful comment that goes to the heart of the matter. As pointed out in the manuscript, a property of attractor states is their robustness and resistance to minor perturbation. This arises from the dynamic interactions of its numerous elements which interact with and are influenced by others in the system. Alterations of the lncRNAs may not by themselves be sufficient to push the system in a different direction towards another attractor state, but along with other ncRNAs (mainly miRNAs and circRNAs), they contribute to the wiring of the dynamic network. The full details of the network dynamics remain to be elucidated, and ongoing research in this area will provide more data to fill in the “mechanistic gaps” that now exist.
Additional information about ncRNAs and their interactions with other RNAs and proteins are included in the manuscript [Page 5].
Reviewer 3 Report
Comments and Suggestions for Authors
Comments to authors:
The manuscript of a research article, which was written by Dr. Amil Shah, is interesting, classifying ncRNAs and discussing their functions that could cause cancer generation. This review article is very useful to understand the background and the presently proposed interplay of ncRNAs, including classical ones, in pathogenesis of cancer. Although this manuscript might be valuable as a general review of ncRNAs, I would suggest author adds more clinical statement, for example what kind of ncRNAs have been identified as etiological factors to cause specific cancers. That could be included in Tables with some appropriate references. If the author anticipates to contribute to developments in cancer treatment, the manuscript would naturally be suitable to be published and read not only by basic scientists but also by medical researchers.
Recommendation: Minor revision
General comments
This review article mainly describes the background of ncRNA and presents some prospects to establish novel treatment of cancer by targeting or introducing ncRNAs. However, if the title is “The primary role of noncoding RNA in the pathogenesis of cancer”, it would be important to be conscious about clinical developments to diagnose or treat specific cancers, as well.
Specific comments
Table 1: Please add some examples that ncRNAs are included in enzyme or the complexes that have enzymatic activities. For instance, telomerase that is highly activated in some specific cancer carries ncRNA, TERC, which serves as a template to elongate chromosomal ends.
Table 2: Some of the example ncRNAs that are thought to be associated with specific types of cancer had better be indicated. Like protein-encoding genes, they would be classified as cancer-generators and cancer-suppressors. Moreover, the author can note the signaling pathways that are regulated by the ncRNAs with the appropriate references.
Minor comments
P1, L2: The title, “The primary role of noncoding RNA in the pathogenesis of cancer”; “The primary roles of noncoding RNAs in the pathogenesis of cancer”?
P2, L83: post-translational modifications; post-translational modifications, including what? Please provide some examples.
P4, L153: lcnRNA; lncRNA
P5, L187 and 190: trans-; trans- should be typed in italic
P5, L182 to 187: Do the references [56] and [57] conflict with each other? I guess some lncRNAs are transcribed bidirectionally with other mRNAs, and others are unidirectional, which might be dependent on the chromosomal structures. Please discuss more about that.
Author Response
Review 3
The manuscript of a research article, which was written by Dr. Amil Shah, is interesting, classifying ncRNAs and discussing their functions that could cause cancer generation. This review article is very useful to understand the background and the presently proposed interplay of ncRNAs, including classical ones, in pathogenesis of cancer. Although this manuscript might be valuable as a general review of ncRNAs, I would suggest author adds more clinical statement, for example what kind of ncRNAs have been identified as etiological factors to cause specific cancers. That could be included in Tables with some appropriate references. If the author anticipates to contribute to developments in cancer treatment, the manuscript would naturally be suitable to be published and read not only by basic scientists but also by medical researchers.
This paper is an overview of the role of ncRNAs in cancer and a follow-up paper of the clinical implications is in preparation. Nonetheless, examples of the interactions of ncRNAs in certain cancer types have been added to provide more clinical perspective [Page 7].
General comments
This review article mainly describes the background of ncRNA and presents some prospects to establish novel treatment of cancer by targeting or introducing ncRNAs. However, if the title is “The primary role of noncoding RNA in the pathogenesis of cancer”, it would be important to be conscious about clinical developments to diagnose or treat specific cancers, as well.
In the general clinical arena, there is some effort to develop ncRNAs as biomarkers, but the concept of ncRNA as a significant player in cancer pathogenesis is not widely appreciated; this paper should address this gap. The potential for developing anticancer therapeutics is included. A statement of the role of ncRNAs as in diagnosis is added to the manuscript.
Specific comments
Table 1: Please add some examples that ncRNAs are included in enzyme or the complexes that have enzymatic activities. For instance, telomerase that is highly activated in some specific cancer carries ncRNA, TERC, which serves as a template to elongate chromosomal ends.
Noted. More information about ncRNA interaction with proteins is added, Specifically, the example of telomerase has been added, as suggested [Page 7].
Table 2: Some of the example ncRNAs that are thought to be associated with specific types of cancer had better be indicated. Like protein-encoding genes, they would be classified as cancer-generators and cancer-suppressors. Moreover, the author can note the signaling pathways that are regulated by the ncRNAs with the appropriate references.
Examples of ncRNA interactions with proteins are added in the text to emphasize this point [Page 7, lines 285-295].
The interaction of ncRNAs with proteins can lead to activation or suppression of well-known cancer drivers, and consequently exert an effect on their signaling pathways. However, the emphasis is on the overarching regulatory role of ncRNAs in the cell from control of gene expression to modulation of protein activity. The principal focus of the paper is to draw attention to this less recognized aspect of cell regulation, whose perturbation might be a major player in tumorigenesis.
(As an aside, the present classification of ncRNAs by size and structure is not particularly useful clinically. A better classification system is required and this will, undoubtedly, emerge as more data are gathered and clinical correlations made.)
Minor comments
P1, L2: The title, “The primary role of noncoding RNA in the pathogenesis of cancer”; “The primary roles of noncoding RNAs in the pathogenesis of cancer”?
The title is “generic”. However, I can understand the use of the plural terms. I have no strong opinion on this point, and I will defer to the editorial office on this change.
P2, L83: post-translational modifications; post-translational modifications, including what? Please provide some examples.
Done. (This refers to breaking or formation of covalent bonds on polypeptide backbones or their amino acid side chains to expand their diversity; this contributes to the emergence of organismal complexity.) [e.g., Zhong Q, et al. Protein posttranslational modifications in health and diseases: Functions, regulatory mechanisms, and therapeutic implications. MedComm (2020). 2023 May 2;4(3):e261. doi: 10.1002/mco2.261. PMID: 37143582; PMCID: PMC10152985.]
P4, L153: lcnRNA; lncRNA
Noted, with thanks.
P5, L187 and 190: trans-; trans- should be typed in italic
Done.
P5, L182 to 187: Do the references [56] and [57] conflict with each other? I guess some lncRNAs are transcribed bidirectionally with other mRNAs, and others are unidirectional, which might be dependent on the chromosomal structures. Please discuss more about that.
The references do indeed describe two different mechanisms of lncRNA biosynthesis. Nojima [2022, ref 56] describes the conventional view, while Wang [2024, ref 57] puts forward a newly discovered mechanism of lncRNA synthesis. This is a new area of inquiry and more details will, undoubtedly, be forthcoming as research progresses.
Round 2
Reviewer 1 Report
Comments and Suggestions for Authors
The review titled "The primary role of noncoding RNA in the pathogenesis of cancer," is enhanced. However, there are some minor weaknesses that need to be addressed before this research article becomes acceptable for publication. In specific, Table 1 provides a useful overview of noncoding RNA types and functions. However, to enhance its informativeness and relevance, the authors should include representative examples for each ncRNA category (e.g., miR-21, HOTAIR, SNORD115) along with appropriate references to support their inclusion.